# The Influence of a Modified 3rd Generation Cementation Technique and Vaccum Mixing of Bone Cement on the Bone Cement Implantation Syndrome (BCIS) in Geriatric Patients with Cemented Hemiarthroplasty for Femoral Neck Fractures

**DOI:** 10.3390/medicina58111587

**Published:** 2022-11-03

**Authors:** Ulf Bökeler, Alissa Bühler, Daphne Eschbach, Christoph Ilies, Ulrich Liener, Tom Knauf

**Affiliations:** 1Department for Orthopaedics and Trauma Surgery, Marienhospital Stuttgart Böheimstrasse 37, 70199 Stuttgart, Germany; 2Center for Orthopaedics and Trauma Surgery, University Hospital Giessen and Marburg, 35039 Marburg, Germany; 3Department for Anesthesia and Intensive Care, Marienhospital Stuttgart, 70199 Stuttgart, Germany

**Keywords:** bone cement implantation syndrome, cementation technique, femoral neck fracture, hip hemiarthroplasty

## Abstract

*Background and Objectives:* Cemented hemi arthroplasty is a common and effective procedure performed to treat femoral neck fractures in elderly patients. The bone cement implantation syndrome (BCIS) is a severe and potentially fatal complication which can be associated with the implantation of a hip prosthesis. The aim of this study was to investigate the influence of a modified cementing technique on the incidence of BCIS. *Material and Methods:* The clinical data of patients which were treated with a cemented hip arthroplasty after the introduction of the modified 3rd generation cementing technique were compared with a matched group of patients who were treated with a 2nd generation cementing technique. The anesthesia charts for all patients were reviewed for the relevant parameters before, during and after cementation. Each patient was classified as having no BCIS (grade 0) or BCIS grade 1,2, or 3 depending on the severity of hypotension, hypoxia loss of consciousness. *Results:* A total of 92 patients with complete data sets could be included in the study. The mean age was 83 years. 43 patients (Group A) were treated with a 2nd and 49 patients (Group B) with a 3rd generation cementing technique. The incidence of BCIS grade 1,2, and 3 was significantly higher (*p* = 0,036) in group A (n = 25; 58%) compared to group B (n = 17; 35%). Early mortality was higher in group A (n = 4) compared to group B (n = 0). *Conclusions:* BCIS is a potentially severe complication with a significant impact on early mortality following cemented hemiarthroplasty of the hip for the treatment of proximal femur fracture. Using a modified 3rd generation cementing technique, it is possible to significantly reduce the incidence of BCIS and its associated mortality.

## 1. Introduction

Femoral neck fractures are common injuries in elderly patients. Despite advances in the medical and surgical management, these injuries are still associated with a peri- and postoperative 30-day mortality of up to 10% [1]. Despite a short modulation of incidence during the COVID epidemic, the total number of fractures is expected to rise significantly in the next decade [2]. In elderly, less mobile patients cemented hemiarthroplasty is the currently recommended treatment of choice for displaced femoral fractures [3]. Although associated with a longer operation time, cemented hemiarthroplasty leads to faster recovery, less pain and better functional results than uncemented fixation of the femoral stem [4,5,6,7,8].

Despite these superior clinical results and a lower revision rate of cemented stems [4,5,6,7] the use of cement still remains controversial [9,10,11], because cemented hip arthroplasty can be associated with the development of bone cement implantation syndrome (BCIS). First described by Donaldson 2009 it is a potentially fatal complication characterized by transient oxygen desaturation, hypotension, acute increase in pulmonary arterial resistance which in severe cases can lead to right ventricular and subsequent life-threatening cardiovascular failure requiring cardiopulmonary resuscitation [12,13,14]. The occurrence of BCIS has resulted in an extensive debate about the fixation of the femoral stem and considerable variation in the practice. Although there is an abundance of literature comparing cemented to uncemented arthroplasty for the treatment of proximal femur fracture [5,7,8,10,15,16,17] there is a scarcity of data available for surgical measures to prevent BCIS. So far, only a single prospective randomized interventional study has analyzed the impact of a modified implantation technique on the incidence and severity of BCIS [18].

The aim of the present study was therefore to investigate the influence of a comprehensive cementing technique on the development of BCIS in a geriatric patient population with hip fractures treated with cemented hemi arthroplasty.

## 2. Material and Methods

After approval by the Institutional review Board, we included all patients into this retrospective study that received a primary cemented hemiarthroplasty due to a femoral neck fracture between January 2007 and December 2015. Between January 2007 and July 2010, a 2nd generation cementing technology was used. Between January 2014 and December 2015, we used a third generation cementing technology. The data collected (from the time of admission to discharge) was extracted from the patient file, the surgical reports and the anesthesia protocols. The following patient characteristics were recorded: age, sex and place of residence. Comorbidities and the severity of pre-existing conditions were determined using the Charlson Comorbidity Index (CCI) [19] and the American Society of Anesthesiologists (ASA) risk classification [20]. In addition, time to surgery, duration of the surgery, type of anesthesia, intraoperative blood pressure, oxygen saturation and adverse perioperative cardiovascular reactions, length of stay, and peri- and postoperative complications were assessed.

The Bone Cement Implantation Syndrome Grades were classified according to Donaldson et al. [12] (Table 1).

Initially, 203 patients with complete data sets were included. Because of inconsistent documentation of the exact time, 111 patients had to be excluded. Depending on the technique of cementation, these patients were divided in two groups. Group A consists of patients who were operated with the second-generation cementing technology, whereas group B had been operated on with a modified third generation cementing technology (Figure 1).

### 2.1. Surgical Technique

All procedures were performed under general anesthesia and monitored by electrocardigram, non-invasive blood pressure management and pulse oximetry. In patients with severe concomitant medical conditions, the monitoring was extended by invasive blood pressure measurement and measurement of the central venous pressure. Prior to cement implantation, the patients were preoxygenated with an FiO2 of 1.0.

In all patients an appropriately sized stem (Bicontact^®^, Aesculap, Tuttlingen, Germany) was cemented with Palacos^®^ (Heraeus Medical, Wehrheim, Germany) through an anterolateral approach in a supine position.

In patients of group A, the 2nd generation cementing technique was used: After femoral canal preparation, a distal cement restrictor was inserted. Then cement was mixed in open atmosphere by hand in an open plastic bowl supplied by the cement manufacturer.

The bone bed was rinsed with a syringe. The cement was applied in a retrograde fashion with the cement gun and the stem was inserted.

In patients of group B, a modified 3rd generation cementing technique was used: After femoral canal preparation, a distal cement restrictor was inserted. The bone bed was thoroughly cleaned with a jet lavage. Cement (Palacos^®^ Heraeus Medical) was mixed with a vacuum mixing system provided by the cement manufacturer (Palamix^®^ Heraeus Medical). Residual blood was removed by lavaging the femoral canal with the jet lavage (Interpulse^®^ Stryker, Kalamazoo, MI, USA). The cement was inserted in a retrograde fashion with a cement gun without a femoral pressurizer.

### 2.2. Statistical Analysis

Statistical analysis was carried out with the SPSS Statistic Program (Version 23, SPSS Inc., Chicago, IL, USA). The level of significance was on α = 0.05 two sided fixed. For comparison of the relative frequency of the BCIS the Chi Quadrat Test, two unpaired tests were used. Further significance analysis of the study results was taken through for nominal variables by the Chi Quadrat Test, for ordinal variables with the Mann-Whitney-U-Tests. For not normally distributed variables, the median was determined. 

## 3. Results

The average age of the 92 patients was 83 years (58–99 years.), 65 were female (71%), and 27 (29%) male (Table 2). The demographic, medical and surgery related date are shown in Table 1. There was no significant difference in the demographic data of the patients.

The proportioning of the patients in the Charlson Comorbidity Index showed a variation in total from 0–9 points with a median of 3 points. There was no difference in the variation of the CCI in between the groups with a median of 3 in both groups.

The majority of the patients (91%) were operated on in the first 48 h following admission. In group B significantly more patients were operated within the first 24 h after admission (Table 2).

The statistical analysis of age, ASA score, CCI and duration of the operation did not reveal a statistical difference between group A and B.

### 3.1. Occurrence of BCIS

42 patients (46%) developed a BCIS. The BCIS occurred significantly (*p* = 0.036) more frequently in Group A (2nd generation cementing technique) (n = 25, 58%) than in Group B (3^rd^ generation cementing technique) (n = 17, 35%). In addition, when BCIS occurred it was more pronounced in patients of Group A (Figure 2).

### 3.2. Complications and Mortality

The postoperative complication rate of all patients was 32% (n = 29). Patients of Group A showed a slightly higher rate of complications (n = 15 (35%)) than patients of Group B (n = 14 (29%)), without being statistically significant. The majority of the complication were medical complications and not directly related to the surgery itself (Table 3).

There were four perioperative deaths in Group A and none in Group B. Three of the four patients that died suffered a BCIS. An overview of al complications distributed to the different groups is shown in Table 3. 

The likelihood of complications was increased if BCIS occurred with 15 complications (36%) in BCIS positive group compared to 14 complications (28%) in the BCIS negative group (Table 4). 

## 4. Discussion

Analysis of large registries has demonstrated a high revision rate following uncemented arthroplasty for hip fracture due to periprosthetic femur fractures in [7,21]. Because of the aforementioned scientific data, the NICE guidelines and the National Hip Fracture Data Base in Great Britain use cemented fixation of the femoral component as a marker of quality for patients with femoral neck fractures [22]. However cemented arthroplasty can be associated with Bone Cement Implantation Syndrome (BCIS). First described and classified by Donaldson, it has a wide spectrum of clinical features that range from transient oxygen desaturation, hypotension to cardiovascular collapse requiring cardiopulmonary resuscitation [12]. The clinical features typically occur at a time of cementation but can also occur at the insertion of an uncemented stem.

The incidence of BCIS varies between 19% and 53% [18,23,24]. It is likely that the real incidence is higher because mild reactions with a blood pressure drop below 20% are not classified as BCIS. The distribution of the severity of BCIS in our study is comparable to the data from Olsen et al. [25]. The high incidences of BCIS do document that it is a potential problem for patients suffering from a proximal femur fracture. A recent analysis documented a trend towards excess mortality in a subgroup of ASA class IV patients [26]. Therefore, identification of frail the patients with poor cardiorespiratory reserves and who are most likely to be affected is paramount in the prevention of BCIS.

Since in-hospital mortality is significantly increased if BCIS occurs, several anesthesiology and surgical measures have been recommended to decrease the incidence of BCIS [23]. The surgical measures include the routine insertion of intramedullary plugs, the placement of medullary drains during cement insertion or alternatively a femoral bore hole and the placement of the patients in a lateral decubitus position in the case of pulmonary disease [18,27]. These surgical recommendations are rather based on assumptions deducted from the analysis on cemented vs. uncemented arthroplasties and not from results of interventional studies. 

So far, only a single study has investigated the impact of a modified cementing technique on the development of BCIS. In a prospective randomized study on 72 patients, Leidinger et al. were able to demonstrate a significant reduction in mortality from 14% in the control group and 3% in the group where the bone cement was prepared with a vacuum mixing system. In addition, the rate of echocardiographically diagnosed pulmonary embolism and circulatory insufficiency was significantly decreased. Unfortunately, they did not grade the severity of the BCIS according to Donaldson; therefore, a direct comparison with our study is not possible [18]. In contrast to our study, they did not use an intramedullary plug and pulsatile lavage before implanting the stem. Our study is the first to investigate the impact of a modified 3rd generation cementing technique on the development of BCIS. We were able to demonstrate a significant reduction in the incidence of BCIS when a modified 3rd generation cementation technique was used (35% 3rd generation vs. 58% 2nd generation cementation technique). The 3rd generation cementing technique aims to improve the inter interlock of cement through thorough bone bed preparation with a pulsatile lavage, use of a distal cement restrictor, retrograde cement application and preparation of cement in a vacuum and femoral pressurization [28].

Bone cement consists of two components. Pre- polymerized polymethyl methacrylate (PMMA) which is present as white powder and the liquid monomer of methyl methacrylate (MMA). When mixing the substances, a catalyst initiates the polymerization of the monomer fluid and the PMMA “pearls” are entrapped within the polymerized monomer.

There is compelling evidence that bone cement is an independent risk factor in the development of BCIS. Christie et al. were able to show that the cemented arthroplasty caused greater and more prolonged embolic cascades than did uncemented arthroplasty [29]. The exact mechanism of this cement mediated pulmonary embolism remains unclear because monomer concentrations in the circulation are very low [30].

Vacuum preparation of cement has been shown to reduce micro and macro pores, resulting in improved strength with a lower risk of aseptic loosening [30]. In addition, vacuum bone cement mixing systems have been shown to significantly remove monomer fumes [31]. It has been hypothesized that monomer-mediated vasoconstriction and mediator release, complement activation and endothelial damage caused by cement particles induce vasoconstriction and pulmonary vascular resistance [12,30]. Given the results of our study and that of Leidinger et al. it is highly likely that the removal of monomer fumes has a significant clinical impact because the vacuum cementing technique significantly reduced the incidence of BCIS.

Although the pathophysiology of BCIS still is not completely understood, there is evidence that pulmonary embolism plays a key role in the etiology of BCIS. Embolic material can be detected in 60% of patients with cemented hip replacement compared to 6% of patients with uncemented hip replacement [32]. In addition, Leidinger et al. were able to detect intra pulmonary thrombotic material in all patients with BCIS that were resuscitated [18].

The use of a pulsatile lavage removes debris caused by femoral canal preparation and fatty marrow particles facilitating cement interdigitation. Given the pathophysiology of BCIS, it is highly likely that thorough jet lavage also reduced the number of intramedullary particles that could dislodge into the systemic circulation, resulting in reduced histamine release, complement activation and emboli formation. In addition to pulsatile lavage, we consistently used a distal cement restrictor (plug) in all patients. In contrast to Weingärtner et al., who have advocated against a medullary plug, we believe that placing a medullary plug does reduce the intramedullary volume by creating a “cement compartment” thereby reducing the overall number of intramedullary particles that can potentially dislodge into the circulation. In addition, we further reduced the intramedullary pressure by not using a proximal pressurization device. We were able to show that the addition of pulsatile lavage and a distal cement restrictor to vacuum cementing technique not only reduces the perioperative mortality, but also decreases the incidence of BCIS.

The total complication rate in our study was 32%, which is comparable to other series [10,15], while implant associated complications occurred less frequently than in the literature with 7 % [11,15]. BCIS positive patients showed an increased number of postoperative complications compared to patients who did not develop a BCIS, without being statistically significant. In our study, the incidence of BCIS did not increase with advancing age. This unexcepted finding is difficult to explain because patient related risk factors for the development for BCIS, like age [23], cardiovascular diseases [18,23,33,34] and malignant diseases [35,36] are well described and influence the development and characteristic of a BCIS. Accordingly, in our study, we were not able to determine any significant influence of the ASA score of the patients on developing a BCIS. Although there was an increased incidence of BCIS in patients with ASA 3, we could not find any significance in contrast to the study of Weingärtner et al., which identified the ASA Score as an independent risk factor [23]. Other risk factors, described in the literature are COPD and the long-term medication with warfarin or diuretics [24]. 

The study has several limitations. It is a retrospective study. Not all data could be collected completely, also the documentation accuracy varied. In several cases, the time of the application of cement was not noted in the anaestheological protocol, therefore these patients could not be included in the study. The sample size of both groups was different. However, a comparative analysis of the patients did not reveal any statistically significant differences in the demographic and outcome parameters of the sample subgroups. 

## 5. Conclusions

Although associated with a longer operation time, treatment of proximal femur fractures with cemented hemi or total arthroplasty leads to better functional results and a lower reoperation rate than uncemented fixation of the femoral stem. Nevertheless, BCIS is a potentially severe complication, with a significant impact on early mortality. In our study, we were able to show that using a modified 3rd generation cementing technique, it is possible to significantly reduce the incidence of BCIS and therefore its associated mortality. Further studies are warranted to elucidate the pathophysiology of BCIS.

A rapid response report concerning the perioperative mortality following cemented arthroplasty for hip fractures by the National Patient Safety Agency in Great Britain in 2009 has led to an extensive analysis of a large number of patients from national registries. These revealed an increase in deaths following cemented arthroplasty in the first 24 h following surgery compared to uncemented arthroplasty [37]. However, an analysis of 16.496 patients from British Nation Hip Fracture Data Base documented significantly lower deaths at discharge in patients with cemented arthroplasty [22]. The findings that cemented arthroplasty does not increase the perioperative mortality has been confirmed by a recent meta-analysis [38]. 

In a recent study by Weingärtner et al. on a comparable patient population using vacuum mixing system, 37% of patients developed BCIS [23]. The authors investigated risk factors for the development of BCIS were able to identify age, ASA score and, as the only modifiable parameter, the (non) placement of a distal bore hole as significant risk factors. It is noteworthy that only 28% of the patients did receive a bore hole.

This tendency towards a higher complication rate is clearer presented by the work of Weingärtner et al. [23] who showed a significantly higher rate of cardiovascular complications and a higher in-hospital mortality rate for BCIS positive patients. Work by Olsen et al. demonstrated that with higher BCIS stages, the one year mortality rate is increasing [25].

## Figures and Tables

**Figure 1 medicina-58-01587-f001:**
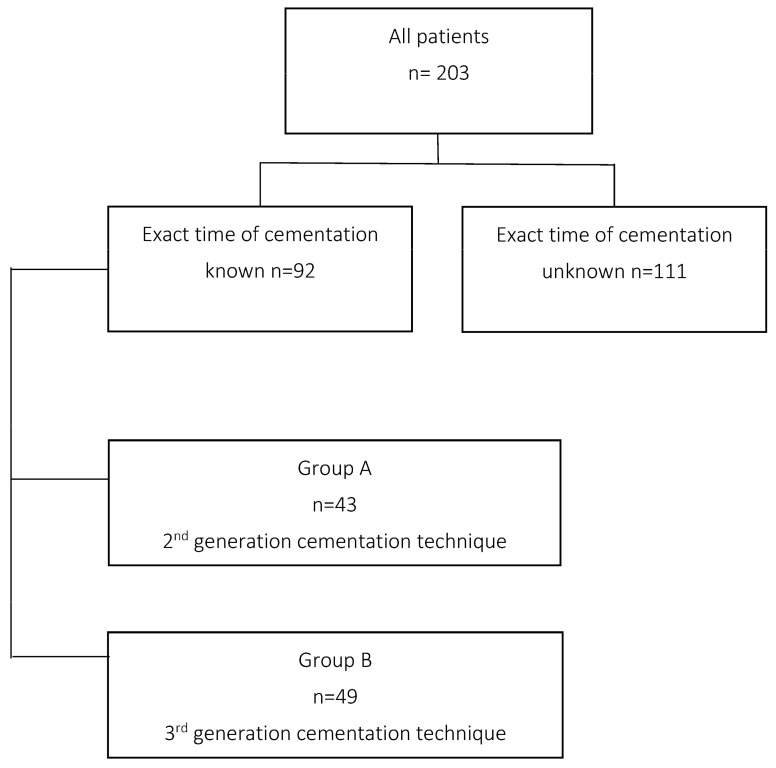
Flowchart.

**Figure 2 medicina-58-01587-f002:**
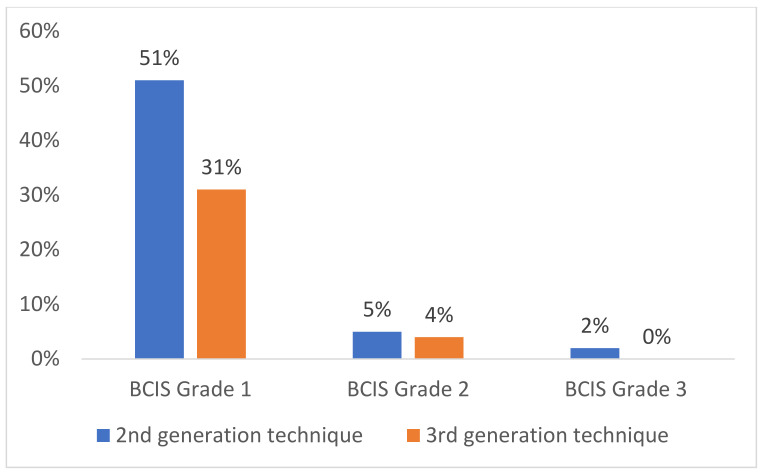
Severity of BCIS in 2nd and 3rd generation technique.

**Table 1 medicina-58-01587-t001:** Classification Bone Cement Implantation Syndrome.

Grade 1	moderate hypoxia (arterial oxygen saturation < 94%) or hypotension (a decrease in systolic arterial pressure (SAP) > 20%)
Grade 2	severe hypoxia (arterial oxygen saturation < 88%) or hypotension (a decrease in SAP > 40%) or unexpected loss of consciousness
Grade 3	cardiovascular collapse requiring cardiopulmonary resuscitation

**Table 2 medicina-58-01587-t002:** Patients characteristics, * *p* < 0.05 (significant).

	Group A	Group B
Age (median)	81	84
Gender n (%)		
- Female	30 (70%)	35 (71%)
- Male	13 (30%)	14 (29%)
Residence n (%)		
- Independent	22 (51%)	30 61%)
- Nursing home	9 (21%)	16 (33%)
- Hospital or rehab clinic	5 (12%)	2 (4%)
- Unknown	7 (16%)	1 (2%)
ASA Grade n (%)		
- ASA 1	0	0
- ASA 2	7 (16%)	14 (29%)
- ASA 3	30 (70%)	33 (67%)
- ASA 4	6 (14%)	2 (4%)
CCI (median)	3	3
Time to surgery in hours n (%)		
- <24	28 (65%) *	42 (86%) *
- 24–48	9 (21%)	5 (11%)
- 48–72	2 (5%)	2 (4%)
- >72	4 (9%)	0

**Table 3 medicina-58-01587-t003:** Overview of complications.

Postoperative Complications	Group A	Group B
	n (%)	n (%)
Overall complications	15 (35%)	14 (29%)
Hip Dislocation	0	1 (2%)
Surgical Site Infection	3 (7%)	2 (4%)
Haematoma	2 (5%)	0
Cardiopulmonary	5 (12%)	5 (10%)
Pulmonary embolism	0	0
Renal failure	0	2 (4%)
Delirium	1 (2%)	4 (8%)
Mortality	4 (10%)	0

**Table 4 medicina-58-01587-t004:** Postoperative Complications in BCIS positive and negative patients.

Postoperative Complications	BCIS Positive	BCIS Negative
	n (%)	n (%)
Number of patients	42 (100%)	50 (100%)
Overall complications	15 (36%)	14 (28%)
Hip Dislocation	0	1 (2%)
Infection	3 (7%)	2 (4%)
Hematoma	1 (2%)	1 (2%)
Cardiopulmonary	5 (12%)	5 (10%)
Pulmonary embolism	0	0
Renal failure	1 (2%)	1 (2%)
Delir	2 (5%)	3 (6%)
Death	3 (7%)	1 (2%)

## Data Availability

The datasets used and/or analyzed during the present study are available from the corresponding author on reasonable request.

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
