# Peer review of "The Influence of a Modified 3rd Generation Cementation Technique and Vaccum Mixing of Bone Cement on the Bone Cement Implantation Syndrome (BCIS) in Geriatric Patients with Cemented Hemiarthroplasty for Femoral Neck Fractures"

_medicina, 2022, doi:10.3390/medicina58111587_

Round 1

Reviewer 1 Report

1.      Its existing title's capitalization should be updated to follow the MDPI format.

2.      Provide all of the author’s emails following MDPI format.

3.      Rearrange the keywords so that they are in alphabetical order.

4.      Make the each of keywords with lowercase font following MDPI format, revise it.

5.      It is unclear whether the author's something new in this work. According to evaluation, several published studies by other researchers in the past adequately explain the issues you made in the present paper. Please be careful to highlight in the introduction section anything really innovative in this work.

6.      In order to highlight the gaps in the literature that the most recent research aims to fill, it is crucial to review the benefits, novelty, and limitations of earlier studies in the introduction.

7.      Since the introduction is too minimal, The Reviewer suggest to explain effort to minimizing revision surgery with computational study. In addition, to support this explanation, the suggested reference should be included as follows: Ammarullah, M. I.; Santoso, G.; Sugiharto, S.; Supriyono, T.; Kurdi, O.; Tauviqirrahman, M.; Winarni, T. I.; Jamari, J. Tresca Stress Study of CoCrMo-on-CoCrMo Bearings Based on Body Mass Index Using 2D Computational Model. Jurnal Tribologi 2022, 33, 31–8. https://jurnaltribologi.mytribos.org/v33/JT-33-31-38.pdf

8.      To help the reader grasp the study's workflow more easily, the authors could include more visuals to the materials and methods section in the form of figures rather than sticking with the text that now predominates.

9.      Other information about the tool, such as the manufacturer, country, and specifications, should be provided.

10.   What is the basis for patientt selection? Is there any protocol, standard, or basis that has been followed? It is unclear since the patient is very heterogeneous with a small number. The resonance involved impacts the present result makes this study flaws. One major reason for rejecting this paper.

11.   The revised manuscript after peer review must provide detailed information on the error and tolerance of the experimental equipment utilized in this study. Due to the disparate outcomes of other researchers' subsequent studies, it would make for a valuable discussion.

12.   Outcomes must be compared to similar past research.

13.   Explain the present study limitation.

14.   Conclusion section is missing, please provide it.

15.   In the conclusion, please explain the further research.

16.   The reference should be enriched with literature from the last five years. Literature published by MDPI is strongly recommended.

17.   Throughout the manuscript, the authors created paragraphs that were only one or two phrases long, making the explanation difficult to understand. The authors should expand on their explanation to make it a more thorough paragraph. It is advised that one paragraph have at least three sentences, with one sentence functioning as the primary sentence and the other sentences functioning as supporting sentences.

18.   Because of grammatical faults and linguistic style, the authors must proofread the document. MDPI English editing service would be a solution.

19.   Please be aware that the authors followed the MDPI format correctly; modify the current form and recheck, as well as any other problems that have been highlighted.

Author Response

Thank you very much for the thorough review of the paper. We have thoroughly revised the manuscript and addressed all comments.

Reviewer 2 Report

Dear Authors, the topic is estremely interesting in fact the relationship between cementation techique in hip arthroplasty and mortality is very actual

As regards the introduction i suggest to improve this section citing the following article in which Authors underlined the relationship between femur fractures and Covid 19. In fact  it will be actually to introduce the problem also in this pandemic period for Covid 19 citing a general article on epidemiology of femur fractures and covid and mortality .

-The epidemiology of proximal femur fractures during covid-19 emergency in italy: A multicentric study

Ciatti C. et al

Acta BiomedicaVolume 92, Issue 53 November 2021 

-Direct Anterior versus Lateral Approach for Femoral Neck Fracture: Role in COVID-19 Disease

Maccagnano G. et al.

Journal of Clinical MedicineOpen Access Volume 11, Issue 16 August 2022

As regards the M&M and results the section is well described 

As regards the discussion and conclusions, are balanced and well supported by analysis . 

Author Response

Thank you very for your kind response and review. We included your very interesting paper (The epidemiology of proximal femur fractures during covid-19 emergency in italy: A multicentric study) into the introduction. The other paper (Direct Anterior versus Lateral Approach for Femoral Neck Fracture: Role in COVID-19 Disease) we integrated into the discussion. 

Again thank you very much. Many greetings. 

The revised manuscript is attached. 
